# DEEP LOW RANK PROJECTOR FOR KV CACHE COMPRESSION

## ABSTRACT

Large Language Models (LLMs) have become integral to a wide range of natural language processing tasks. A key component enabling fast autoregressive inference in LLMs is the Key-Value (KV) cache, which stores hidden states across decoding steps. However, the KV cache imposes substantial memory overhead, especially in long-context generation. While recent studies have proposed various compression techniques to mitigate this issue, they largely overlook the interaction between the techniques Parameter-Efficient Fine-Tuning (PEFT) methods such as LoRA, under which models are significantly more sensitive to KV cache compression. To address this issue, we propose the Deep Low-Rank Projector (DLRP), a novel adapter that compresses the KV cache along the head dimension while preserving downstream performance in PEFT-adapted models. We introduce the Deep Linear Projector (DLP), which is realized leveraging a Deep Linear Network (DLN). We also propose a novel regularizer that approximates the nuclear norm of the DLP, thereby promoting low-rank structure in the learned projection. After training with the proposed regularizer, we inspect the singular-value spectrum and select the minimum rank satisfying a predefined energy threshold, yielding a compact head dimension that balances compression and accuracy. Based on this rank, we construct the DLRP, fine-tune it on the target task, and merge its factorized layers into a single linear operator for efficient inference. Empirical evaluation confirms that DLRP achieves substantial KV cache compression while maintaining strong performance across diverse LLM benchmarks, offering a practical solution for deploying PEFT-adapted models in memory-constrained settings.

Large Language Models (LLMs), such as the GPT series (Achiam et al., 2023; Brown et al., 2020; Ouyang et al., 2022), Claude series (Enis & Hopkins, 2024), and Qwen series (Yang et al., 2025), have become indispensable for a wide range of downstream natural language processing (NLP) tasks, including text generation (Brown et al., 2020; Raffel et al., 2020), summarization (Pu et al., 2023; Zhang et al., 2024), and code generation (Roziere et al., 2023). A key architectural innovation enabling efficient autoregressive inference in LLMs is the Key-Value (KV) cache (Pope et al., 2023), which retains hidden states from previous decoding steps to avoid redundant computation.

While KV caching dramatically speed up inference, its memory footprint grows linearly with both sequence length and model size, leading to substantial memory consumptions (Shi et al., 2024). To address this issue, recent studies have explored a variety of KV cache compression strategies along different structural axes, including the number of layers (Brandon et al., 2024; Sun et al., 2024; Wu & Tu, 2024; Zuhri et al., 2024; Liao & Vargas, 2024; Goldstein et al., 2024), number of heads (Shazeer, 2019; Ainslie et al., 2023; Yu et al., 2024; Chen et al., 2024), sequence length (Devoto et al., 2024; Wang et al., 2024; Xiao et al., 2023; Li et al., 2024; Cai et al., 2024; Feng et al., 2024; Park et al., 2025), and the dimensionality of the head (Liu et al., 2024; Saxena et al., 2024; Yu et al., 2024; Lin et al., 2024).

However, a common limitation of these approaches is that they are primarily developed and evaluated their method under the base models (e.g., pretrained or instruction-tuned) without accounting for downstream adaptation phase. In practice, models are frequently fine-tuned using Parameter-Efficient Fine-Tuning (PEFT) methods such as LoRA (Hu et al., 2022), which introduce small trainable modules while keeping the base model frozen. Notably, we observe that these PEFT-adapted models are considerably more sensitive to KV cache compression. As Figure 1 shows, compressing the KV cache leads to significant performance drops in LoRA-adapted models compared to their

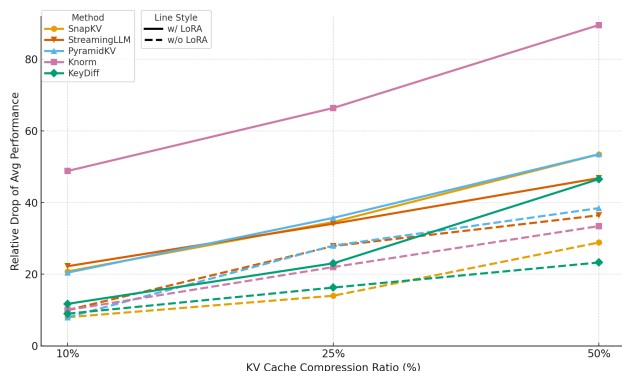

Figure 1: Relative drop of average performance across five benchmark tasks (GSM8K, PIQA, HellaSwag, XSum, and CNN/DM) under different KV cache compression rates (10%, 25%, 50%). *Relative Drop of Average Performance* denotes the percentage reduction in the task-averaged performance relative to the baseline without KV cache compression. LoRA-dapted models (solid lines) exhibit larger performance degradation than non-adapted models (dashed lines), indicating greater sensitivity to compression.

non-adapted counterparts. This indicates a crucial interaction between PEFT mechanisms and KV cache compression that existing methods fail to address.

To mitigate this issue, we introduce the Deep Low-Rank Projector (DLRP), a novel adapter that compresses the KV cache along the head dimension while enabling effective adaptation to downstream tasks. First, we define the Deep Linear Projector (DLP) as a Deep Linear Network (DLN)—a composition of purely linear layers. While DLNs lack non-linear activations, their compositional depth allows them to approximate the optimization and generalization behavior of non-linear architectures such as MLPs. This enables the DLP to effectively capture task-relevant information despite the structural simplicity of linear transformations. To guide the DLP toward a compact, task-aligned projection, we propose a novel regularizer that theoretically approximates the nuclear norm of the DLP. This regularizer elicits the KV cache to be represented in a low-dimensional (i.e., low-rank) subspace that retains task-relevant information. After regularized training, we examine the singular-value spectrum of the learned DLP and select the smallest rank that satisfies a predefined energy threshold, yielding a task-preferred compression level that balances compactness and performance. Based on this rank, we instantiate a Deep Low Rank Projector (DLPR) as a DLN whose output dimension equals the selected rank. We fine-tune the DLRP on the downstream task and then fold its factorized layers into a single linear matrix, which enables efficient inference while preserving compatibility with lightweight adaptation workflows. This process explicitly captures the interaction between PEFT mechanisms and KV cache compression by optimizing the adapter within the learned low-rank subspace induced by regularized training. Empirically, we show that the DLRP not only achieves substantial KV cache compression but also preserves performance under PEFT across a wide range of LLM benchmarks. Our contributions can be summarized as follows:

- We propose Deep Low-Rank Projector (DLRP), a novel adapter that compresses the KV cache along its head dimension while preserving its ability to adapt to downstream tasks.

- The DLRP is built on a deep linear network (DLN) – a stack of purely linear layers – marking a novel application of DLNs as adapter modules in large language models.

- We propose a new regularizer that approximates the nuclear norm of the Deep Linear Projector (DLP), encouraging low-rank projections to compress the KV cache.

- DLRP consistently outperforms existing KV Cache compression methods across a range of benchmarks with superior trade-offs between compression rate and task performance.

# 1 RELATED WORKS

## 1.1 KEY-VALUE (KV) CACHE COMPRESSION

Key-Value (KV) Cache stores tensors of the shape (number of layers, number of heads, sequence length, feature dimension). Prior works have proposed the KV cache compression method along various axes, including the number of layers (Brandon et al., 2024; Sun et al., 2024; Wu & Tu, 2024; Zuhri et al., 2024; Liao & Vargas, 2024; Goldstein et al., 2024), number of heads (Shazeer, 2019; Ainslie et al., 2023; Yu et al., 2024; Chen et al., 2024), sequence length (Devoto et al., 2024; Wang et al., 2024; Xiao et al., 2023; Li et al., 2024; Cai et al., 2024; Feng et al., 2024; Park et al., 2025), and feature dimensionality (Liu et al., 2024; Saxena et al., 2024; Yu et al., 2024; Lin et al., 2024). YOCO (Sun et al., 2024) introduces a dual-decoder architecture in which only the self-decoder encodes global key-value caches, while the cross-decoder reuses them via cross-attention, reducing KV cache size proportionally to the number of cross-decoder layers. SnapKV (Li et al., 2024) compresses the cache by storing less important key-value pairs in low precision while preserving critical ones in high precision. MatryoshkaKV (Lin et al., 2024) reduces the feature dimension of KV caches through trainable orthogonal projection matrices. While these methods effectively reduce KV cache size in base models, they do not consider parameter-efficient fine-tuning (PEFT) scenarios. In this work, we propose a novel adapter that enables efficient KV cache compression along the feature dimension, while preserving performance under PEFT settings.

## 1.2 DEEP LINEAR NETWORKS

Deep Linear Networks have been widely used as analytical tools to understand the behavior of nonlinear neural networks. Even in the depth-0 cases–equivalent to linear regression–DLNs offer valuable insights into generalization in over-parapmeterized regimes (Ziyin et al., 2022; Hastie et al., 2022). The training dynamics of a depth-1 DLNs have also been studied to shed light on the dynamics of learning in nonlinear neural networks (Saxe et al., 2013). While DLNs are as expressive as linear models, the introduction of depth induces non-trivial loss landscapes, which are often considered useful proxies for those of nonlinear neural networks (Kawaguchi, 2016; Hardt & Ma, 2016; Laurent & Brecht, 2018). DLNs continue to provide theoretical insights into modern deep learning problems, such as posterior collapse in variational autoencoders (Lucas et al., 2019; Wang & Ziyin, 2022) and neural collapse in contrastive learning (Tian, 2022). Inspired by the view that DLNs can mimic the behavior of nonlinear neural networks, our work is the first to leverage DLNs not just as theoretical tools but as functional building blocks for desiginig an adapter that enables both KV cache compression and parameter-efficient fine-tuning.

# 2 BACKGROUNDS

## 2.1 KEY-VALUE (KV) CACHE IN TRANSFORMER

In an $L$-layer transformer, the new token embedding $x_t \in \mathbb{R}^{1 \times d_{\text{embed}}}$ at decoding step $t$ and $l$-th layer is used to compute the query vector $q_l^t$, key vector $k_l^t$, and value vector $v_l^t$ as follows:

$$q_l^t = x_t W_{Q_l}, \quad k_l^t = x_t W_{K_l}, \quad v_l^t = x_t W_{V_l}, \tag{1}$$

where $W_{Q_l} \in \mathbb{R}^{d_{\text{embed}} \times d_{\text{head}}}, W_{K_l} \in \mathbb{R}^{d_{\text{embed}} \times d_{\text{head}}}, W_{V_l} \in \mathbb{R}^{d_{\text{embed}} \times d_{\text{head}}}$ are query, key, and value matrices, respectively. The newly computed $k_l^t$ and $v_l^t$ are then appended to the cached key and value matrices from previous steps:

$$K_l^t = \text{Concat}(K_l^{t-1}, k_l^t), \ V_l^t = \text{Concat}(V_l^{t-1}, v_l^t) \in \mathbb{R}^{t \times d_{\text{head}}}, \tag{2}$$

where $K_l^{t-1} \in \mathbb{R}^{(t-1) \times d_{\text{head}}}$ and $V_l^{t-1} \in \mathbb{R}^{(t-1) \times d_{\text{head}}}$ denote the cached key and value matrices of tokens for the previous tokens $x_1, \cdots, x_{t-1}$. These cached matrices are subsequently used in the scaled dot-product attention computation for token $x_t$. The attention output $z_l^t \in \mathbb{R}^{1 \times d_{\text{head}}}$ for the token $x_t$ at step $t$ is computed as:

$$z_l^t = \text{Softmax}\left(\frac{q_l^t K_l^{t\top}}{\sqrt{d_k}}\right) V_l^t \tag{3}$$

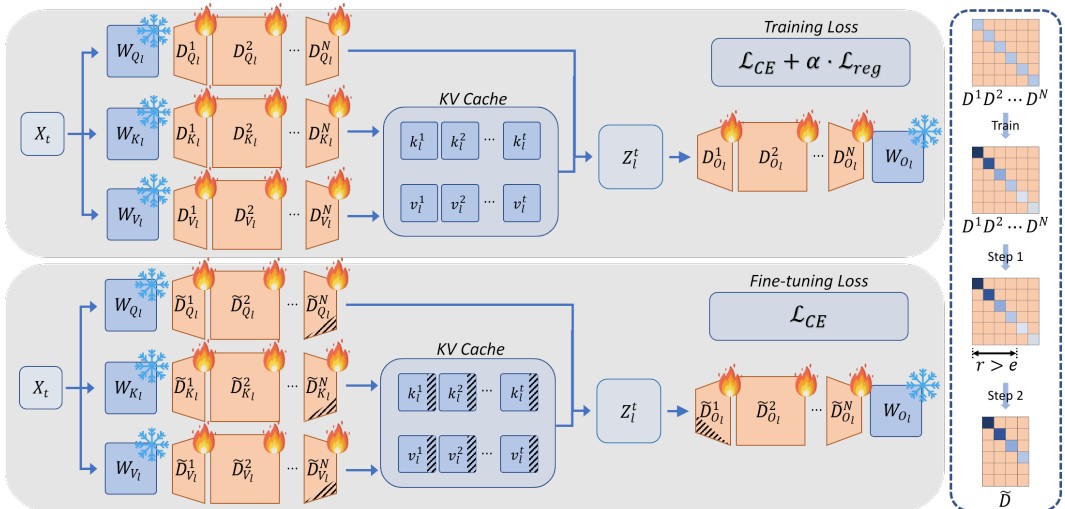

Figure 2: Visualization of the overall framework. The top diagram shows the training stage. For training, Deep Linear Projectors (DLPs) are attached to the frozen backbone network (shown with snowflake).The regularization term $\mathcal{L}_{\text{Reg}}$ encourages DLPs (marked with fire) to learn low-rank structure while preserving task performance. The right-hand figure with dotted line depicts the change in DLP. DLPs gradually obtain low-rank form during training. After training, through step 1, we select rank $r$ that satisfies $\sum_{i=1}^{r} \sigma_i^2 \geq e \sum_{i=1}^{d_{\text{head}}} \sigma_i^2$, where $e$ is the energy preservation threshold. During step 2, using the rank, we construct corresponding Deep Low-Rank Projections (DLRPs), denoted $\tilde{D}$. Finally, as shown in the bottom figure, constructed $\tilde{D}$ are trained for inference.

This key and value reuse process can be applied across the different attention heads within each layer of the transformer. As a result, the KV cache stores tensors of shape `[number of layers, number of heads, sequence length, head dimension]`.

## 2.2 Deep Linear Networks (DLNs)

Deep Linear Networks (DLNs) are fundamental model for studying optimization and generalization in deep learning. This model is a fully connected neural network that excludes intermediate non-linearities (e.g., ReLU). Formally, an DLN $f$ with $L$-layer is defined as:

$$f(x; \theta) = W_L W_{L-1} \cdots W_1 x, \quad \theta = \{W_1, \cdots, W_L\}, \tag{4}$$

where $W_l \in \mathbb{R}^{d_l \times d_{l-1}}$ is a weight matrix of $l$-th layer, and $x \in \mathbb{R}^{d_1}$ denote the input. While DLNs do not gain expressiveness from depth—since they implement only linear input-output mappings—they nonetheless exhibit optimization and generalization behaviors that resemble several characteristics of MLPs. For example, they induce highly non-convex training objectives with numerous minima and saddle points Ge et al. (2015); Lee et al. (2016), and exhibit an implicit bias toward low-rank solutions under gradient descent Li et al. (2020); Huh et al. (2021).

## 3 Method

In this section, we introduce a novel adapter that compresses the KV cache effectively. The overall workflow of proposed method is illustrated in Figure 2[1]. We begin by introducing the Deep Linear Projector (DLP), along with a novel regularizer that effectively enforces a low-rank structure on the projector. Next, we describe how to extract an appropriate low rank from the trained DLP and use it to construct Deep Low-Rank Projector (DLRP). Finally, we explain how the DLRP is fine-tuned and constructed as a lightweight adapter module. The algorithms used for regularized training, fine-tuning, and inference deployment are provided in Appendix A.

---

[1]Snowflake and Fire icons created by Freepik–Flaticon

## 3.1 DEEP LINEAR PROJECTOR (DLP)

For the query matrix $W_{Q_l} \in \mathbb{R}^{d_{\text{embed}} \times d_{\text{head}}}$ in the $l$-th transformer layer, we first define the Deep Linear Projector (DLP) as follows:

$$D_{Q_l}^1 D_{Q_l}^2 \cdots D_{Q_l}^N \in \mathbb{R}^{d_{\text{head}} \times d_{\text{head}}}, \tag{5}$$

where the matrices at each end $D_{Q_l}^1 \in \mathbb{R}^{d_{\text{head}} \times d_{\text{hidden}}}$, $D_{Q_l}^N \in \mathbb{R}^{d_{\text{hidden}} \times d_{\text{head}}}$, and intermediate matrix $D_{Q_l}^n \in \mathbb{R}^{d_{\text{hidden}} \times d_{\text{hidden}}}$ for $n = 2, \cdots, N-1$. Here, $d_{\text{hidden}}$ denotes the hidden dimension used within the DLP. Analogously, we define the DLPs for the key, value, and output matrices as:

$$\begin{aligned} D_{K_l}^1 D_{K_l}^2 \cdots D_{K_l}^N &\in \mathbb{R}^{d_{\text{head}} \times d_{\text{head}}} \\ D_{V_l}^1 D_{V_l}^2 \cdots D_{V_l}^N &\in \mathbb{R}^{d_{\text{head}} \times d_{\text{head}}} \\ D_{O_l}^1 D_{O_l}^2 \cdots D_{O_l}^N &\in \mathbb{R}^{d_{\text{head}} \times d_{\text{head}}} \end{aligned} \tag{6}$$

To encourage low-rankness in these projectors, an established approach is to employ a nuclear norm regularizer (Alvarez & Salzmann, 2017). However, computing this regularizer requires a Singualr Value Decompoistion (SVD), which entails $O(d^3)$ computational cost. To address this limitation, we propose the following novel regularizer:

$$\mathcal{L}_{\text{Reg}} = \sum_{l=1}^{L} \sum_{n=1}^{N} \left( ||D_{Q_l}^n||_F + ||D_{K_l}^n||_F + ||D_{V_l}^n||_F + ||D_{O_l}^n||_F \right) \tag{7}$$

This regularizer function as a practical surrogate of the nuclear norm. Specifically, it exhibits the same low-rank promoting effect while avoiding the costly SVD computation. The precise relationship between the proposed regularizer and the nuclear norm is established by the following theorem:

**Theorem 3.1.** *Let $D^1 \cdots D^N \in \mathbb{R}^{d \times d}$ be a Deep Linear Projector. Then, the following inequality holds:*

$$||D^1 \cdots D^N||_* \leq \frac{1}{N} \cdot \left( \sum_{n=1}^{N} ||D^n||_F \right)^N, \tag{8}$$

*where $||\cdot||_*$ is the nuclear norm and $||\cdot||_F$ is the Frobenius norm.*

We provide the proof of this theorem in Appendix B. The proposed regularizer reduces the time complexity required to inducing low-rankness from $O(d^3)$ to $O(d^2)$, making it scalable for large-scale models. As a result, the total loss used for training the DLPs is given by:

$$\mathcal{L}_{\text{CE}} + \alpha \cdot \mathcal{L}_{\text{Reg}}, \tag{9}$$

where $\mathcal{L}_{\text{CE}}$ denotes the cross-entropy loss and $\alpha$ is a regularization strength. By coupling the task-oriented term $\mathcal{L}_{\text{CE}}$ with our regularizer $\mathcal{L}_{\text{Reg}}$, the resulting objective encourages each DLP to learn a low-rank structure while preserving task performance.

## 3.2 DEEP LOW RANK PROJECTOR (DLRP)

Given the trained DLPs for the query, key, value, and output matrices, regularized according to Eq. (9) to promote low rankness, we construct the Deep Low-Rank Projector (DLRP), which compresses KV cache by projecting the key and value vectors onto reduced-dimensional spaces, in following two steps.

**Step 1. Rank Selection from DLPs** We first fold each trained DLP into a single linear matrix by multiplying its $N$ factors. For example, we set $D_{Q_l} := D_{Q_l}^1 D_{Q_l}^2 \cdots D_{Q_l}^N$ for the query matrix in layer $l$. Then, we determine the rank $r$ for each folded DLP using a predefined energy threshold $e \in (0, 1]$, where $e$ is the fraction of total energy captured by the cumulative singular values; concretely, $r$ is the smallest index satisfying $\sum_{i=1}^{r} \sigma_i^2 \geq e \sum_{i=1}^{d_{\text{head}}} \sigma_i^2$. Because the query and key projections must share the same dimensionality (see Eq. (3)), we first compute the per-projector ranks $r_{\text{query}}^l$ and $r_{\text{key}}^l$ by applying the predefined energy threshold $e$ separately to each folded projectors $D_{Q_l}$ and $D_{K_l}$. We then set a shared rank $r_{\text{qk}}^l := \max\{r_{\text{query}}^l, r_{\text{key}}^l\}$ to satisfy the energy threshold for both query and key. Likewise, we obtain $r_{\text{value}}^l$ and $r_{\text{output}}^l$ from $D_{V_l}$ and $D_{O_l}$ using the same threshold and define $r_{\text{vo}}^l := \max\{r_{\text{value}}^l, r_{\text{output}}^l\}$. These ranks are used to construct the DLRP architecture in the following step.

**Step 2. Construct the DLRPs**    Based on the selected ranks $r_{\text{qk}}^l$ and $r_{\text{vo}}^l$, we define a Deep Low Rank Projectors (DLRPs) for the query, key, value, and output matrices as follows:

$$\tilde{D}_{Q_l}^1 \cdots \tilde{D}_{Q_l}^N \in \mathbb{R}^{d_{\text{head}} \times r_{\text{qk}}^l} \qquad \tilde{D}_{K_l}^1 \cdots \tilde{D}_{K_l}^N \in \mathbb{R}^{d_{\text{head}} \times r_{\text{qk}}^l}$$
$$\tilde{D}_{V_l}^1 \cdots \tilde{D}_{V_l}^N \in \mathbb{R}^{d_{\text{head}} \times r_{\text{vo}}^l} \qquad \tilde{D}_{O_l}^1 \cdots \tilde{D}_{O_l}^N \in \mathbb{R}^{r_{\text{qk}}^l \times d_{\text{head}}} \tag{10}$$

In this construction, we adopt the trained DLP's $d_{\text{hidden}}$ as the DLRP's intermediate size because the ranks $r_{\text{qk}}^l$ and $r_{\text{vo}}^l$ are derived from the DLP's structure, making it appropriate for the DLRP to carry that structure over and maintain architectural consistency.

Since the KV cache tensors have the shape `[number of layers, number of heads, sequence length, head dimension]`, the DLRP can compress the KV cache by reducing the head dimension (i.e., $r_{\text{qk}}^l, r_{\text{vo}}^l << d_{\text{head}}$). Because the head dimension is typically the largest axis and the amount of redundant information varies across tasks and layers, automatically discovering suitable low-rank dimensions $r_{\text{qk}}^l$ and $r_{\text{vo}}^l$ is highly beneficial. Detailed analysis on the task and layer dependent patterns are done in Section 5.1.

### 3.3  Fine-Tuning and Deployment

After constructing the DLRPs, we fine-tune them under a standard supervised objective with the backbone frozen. This procedure is similar to parameter efficient fine-tuning (PEFT) methods such as LoRA (Hu et al., 2022), in which only a small subset of adapter parameters is updated. Concretely, we optimize the factorized DLRPs using the cross-entropy loss $\mathcal{L}_{\text{CE}}$. Once training is complete, we fold each factor chain into a single linear matrix via matrix multiplication. For the DLRP for the query matrix, we fold as follows:

$$\tilde{D}_{Q_l}^1 \cdots \tilde{D}_{Q_l}^N \rightarrow \tilde{D}_{Q_l} \in \mathbb{R}^{d_{\text{head}} \times r_{\text{qk}}^l}. \tag{11}$$

Similarly, the DLRPs for key, value, and output matrices can be folded. This consolidation replaces the chain of linear layers with a single projection during inference, keeping deployment overhead minimal. It also stays compatible with standard KV-cache compression strategies (e.g., a reduced head dimension produces smaller cached key/value tensors) and requires no changes to the serving pipeline.

## 4  Experiments

### 4.1  Experimental Setup

We conduct experiments on three publicly available, open-source instruction-tuned models: Qwen3-4B, Qwen3-8B (Qwen et al., 2025), and Mistral-7B-Instruct-v0.3 (Jiang et al., 2023). These models span a range of parameter scales, which allows us to assess the effectiveness of our method across different model sizes. To access core LLM capabilities spanning short form reasoning and long context summarization, we evaluate our method on five widely used benchmarks: GSM8K (Cobbe et al., 2021), PIQA (Bisk et al., 2020), HellaSwag (Zellers et al., 2019), Xsum (Narayan et al., 2018), and CNN/Daily Mail (Hermann et al., 2015).

We evaluate recent KV cache compression methods using implementations from NVIDIA's KV-Press repository (Jegou et al., 2024): SnapKV (Li et al., 2024), StreamingLLM (Xiao et al., 2023), Knorm (Devoto et al., 2024), PyramidKV (Cai et al., 2024), KeyDiff (Park et al., 2025) in three target compression rates (approximately 10%, 25%, and 50%).

All base models are first fine-tuned with LoRA Hu et al. (2022) (rank 32) under a zero-shot regime. To ensure a fair comparison, we attach adapters only to the attention projections (i.e., query, key, value, and output). Each KV cache baseline is then applied to these LoRA-adapted models so that any performance differences are attributable to the compression method rather than to the adaptation process. We measure performance with EleutherAI's `lm-eval-harness` (Gao et al., 2024) under the same zero-shot protocol across all models and benchmarks.

For DLRP, we use a two-layer DLN (i.e., $N = 2$) with hidden dimension $d_{\text{hidden}} = 256$ and regularization strength $\alpha = 0.01$. Ranks are selected using energy thresholds that correspond approximately to 10%, 25%, and 50% compression rates. All projections $D_{Q_l}^n$, $D_{K_l}^n$, $D_{V_l}^n$, and $D_{O_l}^n$ are

Table 1: Performance under different KV cache compression ratios for each baseline and benchmark. †denotes the method in combined with LoRA (rank 32). The base model is Qwen3-4B. R-1, R-2, and R-L means the ROUGE-1, ROUGE-2, and ROUGE-L, respectively. For each compression rate, the best result is shown in **boldface** and the second-best in underlined text.

| Method | CR | GSM8K Acc | PIQA Acc | HellaSwag Acc | Xsum R-1 | R-2 | R-L | CNN/DM R-1 | R-2 | R-L | Avg. |
|---|---|---|---|---|---|---|---|---|---|---|---|
| LoRA (r=32) | | 73.086 | 86.670 | 91.615 | 43.810 | 19.192 | 34.469 | 44.890 | 21.296 | 31.375 | 49.600 |
| SnapKV† | | 37.528 | 61.153 | 83.621 | 38.529 | 16.373 | 30.122 | 40.116 | 18.389 | 27.764 | 39.288 |
| StreamingLLM† | | 31.387 | 79.181 | 85.087 | 36.740 | 15.655 | 29.043 | 34.159 | 13.221 | 22.623 | 38.566 |
| Knorm† | 10% | 1.137 | 54.026 | 50.500 | 29.154 | 10.403 | 22.705 | 29.551 | 11.185 | 19.784 | 25.383 |
| PyramidKV† | | 38.438 | 61.425 | 83.611 | 38.349 | 17.139 | 30.861 | 39.778 | 18.042 | 27.486 | 39.459 |
| KeyDiff† | | 68.234 | 80.745 | 85.059 | 36.266 | 15.095 | 28.863 | 37.374 | 16.568 | 25.786 | 43.777 |
| DLRP (Ours) | | **69.182** | **81.109** | **86.957** | **41.951** | **18.335** | **32.626** | **42.345** | **20.104** | **29.045** | **46.850** |
| SnapKV† | | 11.221 | 46.736 | 75.530 | 35.835 | 15.930 | 28.852 | 36.708 | 16.001 | 25.207 | 32.447 |
| StreamingLLM† | | 11.827 | 74.126 | 76.522 | 33.313 | 13.758 | 26.245 | 29.256 | 10.063 | 19.055 | 32.685 |
| Knorm† | 25% | 0.379 | 36.453 | 24.283 | 22.511 | 7.411 | 17.433 | 20.741 | 6.502 | 14.285 | 16.666 |
| PyramidKV† | | 11.979 | 47.008 | 74.006 | 35.632 | 15.658 | 28.579 | 35.357 | 14.646 | 24.050 | 31.879 |
| KeyDiff† | | 59.287 | 73.271 | 78.333 | 31.139 | 11.997 | 24.494 | 31.200 | 12.457 | 21.356 | 38.171 |
| DLRP (Ours) | | **65.909** | **79.565** | **82.218** | **39.957** | **17.695** | **31.677** | **38.813** | **18.137** | **27.056** | **44.559** |
| SnapKV† | | 0.531 | 19.260 | 55.803 | 30.861 | 13.015 | 24.568 | 30.952 | 11.811 | 20.914 | 23.079 |
| StreamingLLM† | | 4.701 | 65.066 | 58.760 | 28.601 | 10.820 | 22.372 | 24.089 | 7.131 | 15.803 | 26.371 |
| Knorm† | 50% | 0.015 | 6.474 | 2.295 | 9.988 | 2.826 | 7.768 | 8.640 | 1.840 | 6.796 | 5.183 |
| PyramidKV† | | 0.601 | 19.532 | 56.206 | 30.840 | 12.953 | 24.563 | 30.604 | 11.575 | 20.710 | 23.065 |
| KeyDiff† | | 26.535 | 65.356 | 73.499 | 18.675 | 5.674 | 14.550 | 16.712 | 5.109 | 12.461 | 26.508 |
| DLRP (Ours) | | **59.424** | **75.587** | **78.187** | **34.058** | **15.422** | **28.412** | **36.153** | **16.633** | **25.448** | **41.036** |

Table 2: Performance under different KV cache compression ratios for each baseline and benchmark. †denotes the method in combined with LoRA (rank 32). The base model is Qwen3-8B. R-1, R-2, and R-L means the ROUGE-1, ROUGE-2, and ROUGE-L, respectively. For each compression rate, the best result is shown in **boldface** and the second-best in underlined text.

| Method | CR | GSM8K Acc | PIQA Acc | HellaSwag Acc | Xsum R-1 | R-2 | R-L | CNN/DM R-1 | R-2 | R-L | Avg. |
|---|---|---|---|---|---|---|---|---|---|---|---|
| LoRA (r=32) | | 76.403 | 89.772 | 93.428 | 44.586 | 20.909 | 36.117 | 45.378 | 21.810 | 31.759 | 51.129 |
| SnapKV† | | 41.970 | 69.957 | 87.712 | 40.032 | 18.720 | 32.452 | 40.603 | 18.931 | 28.264 | 42.071 |
| StreamingLLM† | | 32.727 | 82.348 | 87.250 | 38.289 | 17.319 | 30.676 | 34.892 | 13.875 | 23.211 | 40.065 |
| Knorm† | 10% | 2.424 | 78.109 | 61.788 | 31.878 | 12.307 | 25.170 | 35.282 | 14.599 | 23.723 | 31.698 |
| PyramidKV† | | 42.807 | 70.174 | 87.752 | 39.953 | 18.646 | 32.429 | 40.247 | 18.631 | 28.103 | 42.082 |
| KeyDiff† | | 72.423 | 82.630 | 87.230 | 37.745 | 16.588 | 30.336 | 38.526 | 17.334 | 26.633 | 45.494 |
| DLRP (Ours) | | **73.882** | **85.373** | **90.532** | **43.159** | **20.177** | **34.564** | **43.336** | **20.850** | **29.790** | **49.074** |
| SnapKV† | | 15.415 | 53.870 | 84.466 | 37.526 | 17.343 | 30.375 | 37.402 | 16.698 | 25.804 | 35.433 |
| StreamingLLM† | | 12.425 | 76.370 | 83.647 | 34.905 | 15.308 | 27.829 | 29.989 | 10.632 | 19.712 | 34.535 |
| Knorm† | 25% | 1.060 | 68.870 | 39.964 | 25.517 | 9.199 | 20.027 | 28.576 | 10.442 | 18.032 | 24.632 |
| PyramidKV† | | 15.854 | 53.435 | 82.491 | 37.307 | 17.213 | 30.165 | 35.925 | 15.330 | 24.667 | 34.710 |
| KeyDiff† | | 60.757 | 75.957 | 84.051 | 32.328 | 13.112 | 25.750 | 32.950 | 13.625 | 22.567 | 40.122 |
| DLRP (Ours) | | **70.444** | **83.757** | **85.767** | **41.108** | **19.487** | **33.517** | **39.797** | **18.757** | **27.757** | **46.710** |
| SnapKV† | | 0.909 | 18.978 | 60.751 | 32.495 | 14.629 | 26.150 | 31.876 | 12.634 | 21.619 | 24.449 |
| StreamingLLM† | | 4.248 | 69.109 | 63.833 | 30.410 | 12.464 | 24.067 | 24.406 | 7.363 | 15.990 | 27.988 |
| Knorm† | 50% | 0.015 | 41.044 | 1.095 | 12.258 | 3.702 | 9.698 | 13.006 | 3.443 | 9.438 | 10.411 |
| PyramidKV† | | 1.368 | 19.196 | 61.288 | 32.527 | 14.653 | 26.192 | 31.486 | 12.379 | 21.418 | 24.501 |
| KeyDiff† | | 31.060 | 63.870 | 78.548 | 20.675 | 6.662 | 16.187 | 20.396 | 6.641 | 14.612 | 28.739 |
| DLRP (Ours) | | **63.644** | **79.628** | **81.563** | **35.134** | **16.999** | **30.122** | **37.119** | **17.295** | **26.138** | **43.071** |

initialized as identity matrices for every layer $l$ and depth $n$. Additional hyperparameter details are provided in Appendix C

## 4.2 EXPERIMENTAL RESULTS

We present the results on Qwen3-4B and Qwen3-8B across KV cache compression rates of 10%, 25%, and 50%. Additional results for Mistral-7B-Instruct-v0.3, are provided in Appendix D. As shown in Table 1, DLRP consistently outperforms baseline methods–SnapKV, StreamingLLM, Knorm, PyramidKV, and KeyDiff–especially as compression intensifies. Whereas these methods often degrade sharply at moderate and high compression rate, DLRP maintains robust performance on reasoning-heavy tasks (e.g., GSM8K) and on long-context summarization (e.g., CNN/DM). For example, at 10% compression, DLRP achieves 69.182% on GSM8K and 86.975% on HellaSwag, surpassing all baselines; even at 25% and 50%, GSM8K accuracy remains 65.909% and 59.242%, respectively. These results indicate that, within a PEFT setting, DLRP remains robust under aggres-

sive KV-cache compression by optimizing the adapters within the low-rank subspace induced by our training procedure. A similar trend is observed in Table 2, where DLRP demonstrates strong performance on Qwen3-8B across all benchmark and compression rate. Notably, it achieves 73.882% GSM8K accuracy at 10% compression and sustains 63.644% even at 50%, significantly outperforming other methods. These results highlight DLRP's robustness and generalizability across different model backbones and task types, including both reasoning and summarization.

Table 3: Average performance comparison of DLP architectures with varying depth $N$ and hidden dimension $d_{\text{hidden}}$ under $25\%$ KV-cache compression. We employ Qwen3-4B as base model.

|  | $N = 1$ | $N = 2$ | $N = 3$ |
|---|---|---|---|
| $d_{\text{hidden}} = 128$ | 43.597 | 45.191 | 44.266 |
| $d_{\text{hidden}} = 256$ | ✗ | **45.482** | 45.085 |
| $d_{\text{hidden}} = 512$ | ✗ | 44.670 | 43.814 |

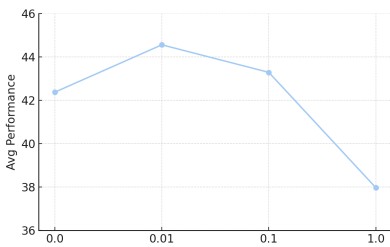

Figure 3: Average performance across five benchmarks under $25\%$ KV-cache compression with varying regularization strength $\alpha$. We employ Qwen3-4B as base model.

### 4.3 ABLATION STUDIES

#### 4.3.1 REGULARIZATION STRENGTH $\alpha$ OF THE REGULARIZER $\mathcal{L}_{\text{REG}}$

We examine the effect of the regularization strength $\alpha$ on DLP training under a 25% KV cache compression target. Figure 3 visualizes the average performance across five benchmarks for each $\alpha \in \{1.0, 0.1, 0.01, 0.0\}$. As $\alpha$ decreases from 1.0 to 0.01, performance improves, indicating that excessively strong regularization limits the projector's expressivity. When $\alpha = 0$, performance drops slightly, suggesting that a modest regularization terms is necessary to reliably identify an appropriate rank (via the energy threshold) for each task while maintaining performance at the target compression level. Based on this finding, we set $\alpha = 0.01$ as the default value for all experiments.

### 4.4 THE WIDTH AND DEPTH OF DEEP LINEAR PROJECTOR

We study how the architecture of the DLP affects the performance by varying its depth ($N \in \{1, 2, 3\}$) and hidden dimensions ($d_{\text{hidden}} \in \{128, 256, 512\}$). All ablations are conducted under a 25% KV cache compression target, and we report the average performance across five benchmarks. For $N = 1$, the DLP reduces to a single linear map with no intermediate layer; consequently $d_{\text{hidden}}$ is not applicable, and we exclude the pairs (1, 256) and (1, 512). Table 3 shows that the configuration with $N = 2$ and $d_{\text{hidden}} = 256$ performs best overall. In theory, adding more layers to a linear network does not increase its expressiveness, since stacked linear layers are still equivalent to a single linear transformation. However, we observe that different depth and width choices still lead to clear differences in performance. Smaller models (e.g., $d_{\text{hidden}} = 128$) tend to perform worse, possibly due to limited flexibility during training. Larger or deeper models (e.g., $d_{\text{hidden}} = 512$ or $N = 3$) do not improve results and can even slightly hurt performance because deeper linear compositions make optimization harder. In particular, increasing depth beyond two layers consistently shows a small drop in accuracy across all widths. These results suggest that, even in linear network, architectural choices affect how well the model trains, rather than how much it can represent. Based on this finding, we choose the $N = 2$, $d_{\text{hidden}} = 256$ configuration as the default DLP architecture for all experiments.

## 5 DISCUSSION

In this section, we discuss the rank pattern arising from the proposed regularizer, with extended discussion in Appendix E.

### 5.1 EMERGENT RANK PATTERNS ACROSS TASKS

To clarify how low-rank structure varies across layers and tasks, we analyze the per-layer ranks $r_{\text{qk}}^l$ and $r_{\text{vo}}^l$ for each task and characterize their emergent patterns across depth. Each rank is determined

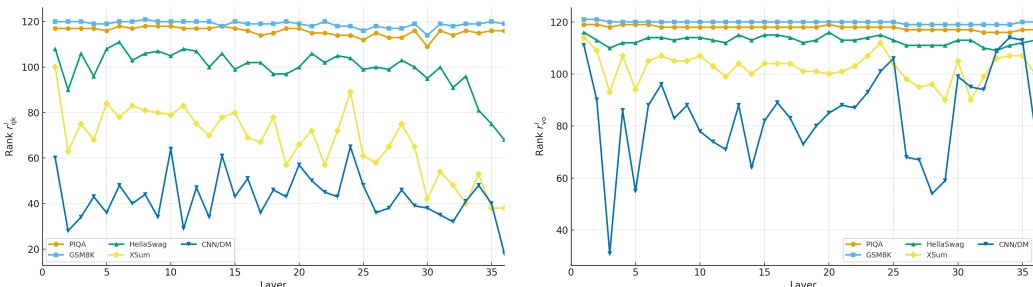

Figure 4: Layer-wise rank pattern for $r_{\text{qk}}$ and $r_{\text{vo}}$. These ranks are determined by an energy threshold $e = 0.95$. We employ Qwen3-4B as base model.

by an energy threshold $e = 0.95$. Figure 4 shows the rank pattern for $r_{\text{qk}}^l$ and $r_{\text{vo}}^l$ revealing clear task-dependent pattern across tasks. Across the five benchmarks, we find that GSM8K and PIQA (short-context, reasoning-intensive) sustain near-full ranks over most layers, indicating strong resistance to compression; HellaSwag exhibits mid-scale ranks consistent with its moderate length and common-sense difficulty; and within summarization, XSum (shorter, more abstractive) induces higher ranks scale than CNN/DM, whose much longer and more redundant inputs yield the lowest rank scale overall. These trends align with established findings in KV-cache compression: redundancy in long contexts permits more aggressive reduction (e.g., via heavy-hitter retention), whereas reasoning-centric workloads are fragile—particularly under prefill compression—and therefore require larger effective subspaces (Xu et al., 2025; Liu et al., 2025). The pronounced layer-wise heterogeneity in the rank pattern further suggests that optimal budgets are non-uniform across depth. Overall, these findings indicate that our regularizer adapts ranks to task demands, preserving capacity where required (reasoning tasks) while collapsing low-energy dimensions for redundant, long-context inputs (CNN/DM). Collectively, these results are consistent with depth-aware compression that funnels information across layers and with adaptive policies that selectively retain impactful tokens/heads.

# 6 LIMITATIONS

One possible limitation of our work is that DLRP focuses solely on compressing the KV-cache's head dimension, leaving the layer count, sequence length, and the number of head unchanged. Although head-dimension compression reliably reduces size regardless of input shape, applying compression to these additional axes could further improve the overall compression rate. Consequently, extensive exploration of combining DLRP with other techniques may yield even greater reductions of the KV cache. This line of research is both valuable and highly relevant to multi-dimensional KV-cache compression. Future work could therefore extend DLRP to operate on multiple cache dimensions and investigate synergistic integrations with existing KV-compression methods.

# 7 CONCLUSION

In this work, we identify a key limitation of KV-cache compression: PEFT-adapted LLMs are unusually brittle under compression. We address this with the Deep Low-Rank Projector (DLRP), which compresses the head dimension via a deep linear projector trained on the downstream objective alongside a nuclear-norm-inspired regularizer that induces low rank. After training, we examine the singular-value spectrum to select the smallest energy-preserving rank, instantiate DLRP at that rank, fine-tune the adapter, and fold its factors into a single linear map for deployment. This procedure optimizes within a learned low-rank subspace and explicitly captures the interaction between PEFT and KV-cache compression. Across diverse LLM benchmarks, DLRP consistently delivers strong memory–accuracy trade-offs and stable adaptation under compression, enabling efficient inference of PEFT-adapted models in memory-constrained settings.

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

# A ALGORITHM

We provide concise algorithms for (1) regularized training the Deep Linear Projector (DLP), (2) fine-tuning the Deep Low-Rank Projector (DLRP), and (3) deploying the DLRP for efficient inference. For brevity, the pseudo-code shows only the attention weights.

---

**Algorithm 1** Regularized Training for the DLP

---

**Require:** Training dataset $\mathcal{D}_{\text{train}}$, Regularization strength $\alpha$
**Require:** Cross-entropy loss $\mathcal{L}_{\text{CE}}$
**Require:** Pre-trained attention weights:

$$W_Q = \{W_{Q_l}\}_{l=1}^L, \quad W_K = \{W_{K_l}\}_{l=1}^L$$
$$W_V = \{W_{V_l}\}_{l=1}^L, \quad W_O = \{W_{O_l}\}_{l=1}^L$$

1: Initialize DLPs:

$$D_Q = \{D_{Q_l}^1 \cdots D_{Q_l}^N\}_{l=1}^L, \quad D_K = \{D_{K_l}^1 \cdots D_{K_l}^N\}_{l=1}^L$$
$$D_V = \{D_{V_l}^1 \cdots D_{V_l}^N\}_{l=1}^L, \quad ; D_O = \{D_{O_l}^1 \cdots D_{O_l}^N\}_{l=1}^L$$

2: Obtain the DLP-applied pre-trained attention weights:

$$W_Q^{\text{DLP}} = \{W_{Q_l} \cdot D_{Q_l}^1 \cdots D_{Q_l}^N\}_{l=1}^L$$
$$W_K^{\text{DLP}} = \{W_{K_l} \cdot D_{K_l}^1 \cdots D_{K_l}^N\}_{l=1}^L$$
$$W_V^{\text{DLP}} = \{W_{V_l} \cdot D_{V_l}^1 \cdots D_{V_l}^N\}_{l=1}^L$$
$$W_O^{\text{DLP}} = \{W_{O_l} \cdot D_{O_l}^1 \cdots D_{O_l}^N\}_{l=1}^L$$

3: **while** not converged **do**
4:   Sample mini-batch $B \sim \mathcal{D}_{\text{train}}$
5:   Compute the cross-entropy loss

$$\mathcal{L}_{\text{CE}}(W_Q^{\text{DLP}}, W_V^{\text{DLP}}, W_K^{\text{DLP}}, W_O^{\text{DLP}}; B)$$

6:   Compute the regularizer $\mathcal{L}_{\text{Reg}}$ via Eq. (7)
7:   Compute the total loss $\mathcal{L}_{\text{total}} = \mathcal{L}_{\text{CE}} + \alpha \cdot \mathcal{L}_{\text{Reg}}$
8:   Update the DLPs ($D_Q, D_K, D_V$, and $D_O$) via the total loss $\mathcal{L}_{\text{total}}$
9: **end while**

---

**Algorithm 2** Fine-Tuning for the DLRP

**Require:** Training dataset $\mathcal{D}_{\text{train}}$, predefined energy threshold $e$
**Require:** Cross-entropy loss $\mathcal{L}_{\text{CE}}$
**Require:** Pre-trained attention weights:

$$W_Q = \{W_{Q_l}\}_{l=1}^L, \quad W_K = \{W_{K_l}\}_{l=1}^L$$
$$W_V = \{W_{V_l}\}_{l=1}^L, \quad W_O = \{W_{O_l}\}_{l=1}^L$$

**Require:** Regularized trained DLPs:

$$D_Q = \{D_{Q_l}^1 \cdots D_{Q_l}^N\}_{l=1}^L; \quad D_K = \{D_{K_l}^1 \cdots D_{K_l}^N\}_{l=1}^L$$
$$D_V = \{D_{V_l}^1 \cdots D_{V_l}^N\}_{l=1}^L; \quad D_O = \{D_{O_l}^1 \cdots D_{O_l}^N\}_{l=1}^L$$

1: **for** $l = 1, \cdots, L$ **do**
2: $\quad r_{\text{query}}^l = \text{rank}(D_{Q_l}^1 \cdots D_{Q_l}^N; e), \quad r_{\text{key}}^l = \text{rank}(D_{K_l}^1 \cdots D_{K_l}^N; e)$
3: $\quad r_{\text{value}}^l = \text{rank}(D_{V_l}^1 \cdots D_{V_l}^N; e), \quad r_{\text{output}}^l = \text{rank}(D_{O_l}^1 \cdots D_{O_l}^N; e)$
4: $\quad r_{\text{qk}}^l = \max(r_{\text{query}}^l, r_{\text{key}}^l), \quad r_{\text{vo}}^l = \max(r_{\text{value}}^l, r_{\text{output}}^l)$
5: **end for**
6: Based on rank $r_{\text{qk}}^l$ and $r_{\text{vo}}^l$, construct the DLRPs via Eq. (10) and initialize them:

$$\tilde{D}_Q = \{\tilde{D}_{Q_l}^1 \cdots \tilde{D}_{Q_l}^N \in \mathbb{R}^{d_{\text{head}} \times r_{\text{qk}}^l}\}_{l=1}^L$$
$$\tilde{D}_K = \{\tilde{D}_{K_l}^1 \cdots \tilde{D}_{K_l}^N \in \mathbb{R}^{d_{\text{head}} \times r_{\text{qk}}^l}\}_{l=1}^L$$
$$\tilde{D}_V = \{\tilde{D}_{V_l}^1 \cdots \tilde{D}_{V_l}^N \in \mathbb{R}^{d_{\text{head}} \times r_{\text{vo}}^l}\}_{l=1}^L$$
$$\tilde{D}_O = \{\tilde{D}_{O_l}^1 \cdots \tilde{D}_{O_l}^N \in \mathbb{R}^{d_{\text{head}} \times r_{\text{vo}}^l}\}_{l=1}^L$$

7: Obtain the DLRP-applied pre-trained attention weights:

$$W_Q^{\text{DLRP}} = \{W_{Q_l} \cdot \tilde{D}_{Q_l}^1 \cdots \tilde{D}_{Q_l}^N\}_{l=1}^L$$
$$W_K^{\text{DLRP}} = \{W_{K_l} \cdot \tilde{D}_{K_l}^1 \cdots \tilde{D}_{K_l}^N\}_{l=1}^L$$
$$W_V^{\text{DLRP}} = \{W_{V_l} \cdot \tilde{D}_{V_l}^1 \cdots \tilde{D}_{V_l}^N\}_{l=1}^L$$
$$W_O^{\text{DLRP}} = \{W_{O_l} \cdot \tilde{D}_{O_l}^1 \cdots \tilde{D}_{O_l}^N\}_{l=1}^L$$

8: **while** not converged **do**
9: $\quad$ Sample mini-batch $B \sim \mathcal{D}_{\text{train}}$
10: $\quad$ Compute the cross-entropy loss

$$\mathcal{L}_{\text{CE}}(W_Q^{\text{DLRP}}, W_V^{\text{DLRP}}, W_K^{\text{DLRP}}, W_O^{\text{DLRP}}; B)$$

11: $\quad$ Update the DLRPs ($\tilde{D}_Q, \tilde{D}_K, \tilde{D}_V$, and $\tilde{D}_O$) via the cross-entropy loss $\mathcal{L}_{\text{CE}}$
12: **end while**

---

**Algorithm 3** Inference for the DLRP (i.e., Deployment)

---

**Require:** Fine-tuned DLRP

$$\tilde{D}_Q = \{\tilde{D}_{Q_l}^1 \cdots \tilde{D}_{Q_l}^N \in \mathbb{R}^{d_{\text{head}} \times r_{\text{qk}}^l}\}_{l=1}^L$$

$$\tilde{D}_K = \{\tilde{D}_{K_l}^1 \cdots \tilde{D}_{K_l}^N \in \mathbb{R}^{d_{\text{head}} \times r_{\text{qk}}^l}\}_{l=1}^L$$

$$\tilde{D}_V = \{\tilde{D}_{V_l}^1 \cdots \tilde{D}_{V_l}^N \in \mathbb{R}^{d_{\text{head}} \times r_{\text{vo}}^l}\}_{l=1}^L$$

$$\tilde{D}_O = \{\tilde{D}_{O_l}^1 \cdots \tilde{D}_{O_l}^N \in \mathbb{R}^{d_{\text{head}} \times r_{\text{vo}}^l}\}_{l=1}^L$$

1: Fold the factorized DLRP into a single linear matrix via Eq. (11):

$$\tilde{D}_Q^{\text{Fold}} = \{\tilde{D}_{Q_l} \in \mathbb{R}^{d_q \times r_{\text{qk}}^l}\}_{l=1}^L$$

$$\tilde{D}_K^{\text{Fold}} = \{\tilde{D}_{K_l} \in \mathbb{R}^{d_q \times r_{\text{qk}}^l}\}_{l=1}^L$$

$$\tilde{D}_V^{\text{Fold}} = \{\tilde{D}_{V_l} \in \mathbb{R}^{d_q \times r_{\text{vo}}^l}\}_{l=1}^L$$

$$\tilde{D}_O^{\text{Fold}} = \{\tilde{D}_{O_l} \in \mathbb{R}^{d_q \times r_{\text{vo}}^l}\}_{l=1}^L$$

2: Deploy the folded DLRPs: $\tilde{D}_Q^{\text{Fold}}$, $\tilde{D}_K^{\text{Fold}}$, $\tilde{D}_V^{\text{Fold}}$, and $\tilde{D}_O^{\text{Fold}}$

---

## B  PROOF

We begin by stating the lemma, followed by the proof of the theorem in Section 3.1.

**Lemma B.1.** *For any two matrices $A \in \mathbb{R}^{m \times n}$, $B \in \mathbb{R}^{n \times p}$, the following inequality holds:*

$$||AB||_F \leq ||A||_F \cdot ||B||_F \tag{12}$$

*where $|| \cdot ||_F$ is the Frobenius norm.*

*Proof.* Let $C = AB \in \mathbb{R}^{m \times p}$. By definition,

$$||C||_F^2 = \sum_{i=1}^{m} \sum_{j=1}^{p} c_{ij}^2 = \sum_{i=1}^{m} \sum_{j=1}^{p} \left( A_i b_j \right)^2, \tag{13}$$

where $c_{ij}$ denotes the $(i, j)$-th entry of $C$, $A_i$ is the $i$-th row of $A$, and $b_j$ is the $j$-th column of $B$. By Cauchy-Schwarz Inequality Bityutskov (2001), we obtain that

$$
\begin{aligned}
||C||_F^2 &\leq \sum_{i=1}^{m} \sum_{j=1}^{p} ||A_i||^2 ||b_j||^2 \\
&= \left( \sum_{i=1}^{m} ||A_i||^2 \right) \left( \sum_{j=1}^{p} ||b_j||^2 \right) \\
&= ||A||_F^2 ||B||_F^2
\end{aligned}
\tag{14}
$$

Taking the quare root of each side completes the proof. $\square$

**Theorem B.1** (Restate). *Let $U^1 \cdots U^N \in \mathbb{R}^{d \times d}$ be a Deep Linear Projector. Then, the following inequality holds:*

$$||U^1 \cdots U^N||_* \leq \frac{1}{N} \cdot \left( \sum_{n=1}^{N} ||U^n||_F \right)^N, \tag{15}$$

*where $|| \cdot ||_*$ is the nuclear norm and $|| \cdot ||_F$ is the Frobenius norm.*

*Proof.* Let $U = U^1 \cdots U^N$ be a deep linear projector and abbreviate $\hat{U} := U^2 \cdots U^N$. By definition, the nuclear norm of $U$ is given by

$$||U||_* = ||U^1 \hat{U}||_* = \text{Tr}(U^1 \hat{U}) \tag{16}$$

By Cauchy-Schwarz Inequality Bityutskov (2001), we can obtain the following inequality:

$$\text{Tr}(U^1 \hat{U}) \leq \sqrt{\text{Tr}\left( (U^1)^T U^1 \right) \text{Tr}\left( \hat{U}^T \hat{U} \right)} \tag{17}$$

By definition, we can calculate the following inequality:

$$\sqrt{\text{Tr}\left( (U^1)^T U^1 \right) \text{Tr}\left( \hat{U}^T \hat{U} \right)} = ||U^1||_F ||\hat{U}||_F \tag{18}$$

By Lemma B.1, we can derive the following:

$$||U^1||_F ||\hat{U}||_F \leq ||U^1||_F \cdots ||U^N||_F \tag{19}$$

Using the AM-GM Inequality, we can derive the following inequality:

$$
\begin{aligned}
||U^1||_F \cdots ||U^N||_F &\leq \frac{1}{N} \left( ||U^1||_F + \cdots + ||U^N||_F \right)^N \\
&= \frac{1}{N} \cdot \left( \sum_{i=1}^{N} ||U^n||_F \right)^N
\end{aligned}
\tag{20}
$$

Combining Eq. (16) - (20) yields the following inequality:

$$||U||_* \leq \frac{1}{N} \cdot \left( \sum_{i=1}^{N} ||U^n||_F \right)^N \tag{21}$$

$\square$

## C  HYPER-PARAMETER SETTING

**Common Setting**  All experiments use AdamW with a warm-up/decay schedule: the learning rate rises linearly to $5 \times 10^{-4}$ over the first $10\%$ of steps, then decays. We adopt a global batch size of 32 and truncate or pad all inputs to 512 tokens across the five datasets.

**Regularized Training (DLP)**  The Deep Linear Projector (DLP) is trained for 10 epochs in bfloat16 on up to four H100 (80 GB) GPUs; this stage completes in under two hours on the full 4-GPU setup.

**Fine-Tuning (DLRP)**  The Deep Low-Rank Projector (DLRP) is fine-tuned for 3 epochs in bfloat16 on up to four H100 (80 GB) GPUs, finishing in less than one hour on four H100 GPUs.

**Fine-Tuning (LoRA)**  The LoRA baseline is likewise fine-tuned for 5 epochs in bfloat16 on up to four H100 GPUs, requiring under one hour to converge.

## D  ADDITIONAL RESULTS

In this section, we report additional results on Mistral-7B-Instruct-v0.3 under KV cache compression rates of 10%, 25%, and 50%. DLRP consistently achieves the highest average performance among the baselines, matching the trends observed on Qwen3-4B and Qwen3-8B, and the advantage becomes more pronounced as compression rate increases.

Table 4: Performance under different KV cache compression ratios for each baseline and benchmark. †denotes the method in combined with LoRA (rank 32). The base model is Mistral-7B-Instruct-v0.3. R-1, R-2, and R-L means the ROUGE-1, ROUGE-2, and ROUGE-L, respectively. For each compression rate, the best result is shown in **boldface** and the second-best in underlined text.

| Method | CR | GSM8K Acc | PIQA Acc | HellaSwag Acc | Xsum R-1 | R-2 | R-L | CNN/DM R-1 | R-2 | R-L | Avg. |
|---|---|---|---|---|---|---|---|---|---|---|---|
| LoRA (r=32) | | 53.030 | 88.152 | 94.604 | 46.366 | 23.174 | 37.693 | 45.482 | 21.766 | 31.945 | 49.135 |
| SnapKV† | | 35.000 | 75.620 | 84.821 | 41.336 | 20.304 | 33.565 | 40.572 | 18.624 | 28.096 | 41.993 |
| StreamingLLM† | | 27.576 | 79.337 | 84.512 | 39.976 | 19.051 | 32.220 | 35.828 | 14.336 | 23.871 | 39.634 |
| Knorm† | 10% | 13.182 | 43.924 | 30.512 | 39.369 | 19.491 | 32.587 | 40.554 | 18.643 | 27.777 | 29.560 |
| PyramidKV† | | 36.667 | 75.717 | 82.929 | 41.262 | 20.461 | 33.504 | 40.327 | 18.452 | 27.934 | 41.917 |
| KeyDiff† | | 45.955 | 78.946 | 84.817 | 41.214 | 20.446 | 33.653 | 40.853 | 19.329 | 28.566 | 43.753 |
| DLRP (Ours) | | **50.742** | **83.176** | **90.730** | **44.637** | **22.251** | **35.876** | **43.174** | **20.678** | **29.771** | **46.781** |
| SnapKV† | | 15.909 | 61.717 | 78.094 | 36.702 | 18.570 | 30.994 | 37.304 | 16.337 | 25.541 | 35.685 |
| StreamingLLM† | | 8.939 | 74.098 | 79.279 | 36.891 | 17.233 | 29.609 | 30.869 | 11.060 | 20.182 | 34.240 |
| Knorm† | 25% | 4.848 | 41.484 | 25.468 | 36.272 | 17.458 | 30.038 | 37.668 | 16.972 | 25.809 | 26.224 |
| PyramidKV† | | 15.455 | 62.457 | 77.502 | 37.568 | 18.575 | 30.210 | 36.824 | 15.925 | 25.202 | 35.524 |
| KeyDiff† | | 37.992 | 74.560 | 79.512 | 38.928 | 19.390 | 31.744 | 38.501 | 18.051 | 26.962 | 40.627 |
| DLRP (Ours) | | **48.371** | **81.567** | **85.867** | **42.520** | **21.482** | **34.808** | **39.610** | **18.621** | **27.728** | **44.508** |
| SnapKV† | | 1.364 | 29.826 | 58.765 | 33.626 | 16.139 | 28.549 | 32.423 | 12.778 | 21.712 | 26.131 |
| StreamingLLM† | | 4.091 | 69.130 | 56.233 | 31.811 | 14.119 | 25.570 | 25.603 | 8.084 | 16.693 | 27.926 |
| Knorm† | 50% | 1.970 | 38.783 | 25.468 | 33.074 | 15.735 | 27.379 | 32.619 | 12.924 | 21.881 | 23.315 |
| PyramidKV† | | 1.515 | 30.435 | 59.180 | 35.335 | 16.958 | 28.585 | 32.295 | 12.621 | 21.666 | 26.510 |
| KeyDiff† | | 17.644 | 63.478 | 72.356 | 35.235 | 16.722 | 28.306 | 35.153 | 15.813 | 24.276 | 34.331 |
| DLRP (Ours) | | **43.660** | **77.516** | **81.662** | **36.295** | **18.734** | **31.257** | **36.915** | **17.130** | **26.099** | **41.030** |

# E ADDITIONAL DICUSSION: DLRP-INDUCED STRUCTURED LOTTERY-TICKET SUBSPACES IN THE KV CACHE

Our analysis of the rank pattern in Figure 4 indicates that DLRP adopts task-adaptive KV cache compression via the per-layer ranks $r_{qk}^l$ and $r_{vo}^l$. On reasoning-intensive benchmarks such as GSM8K and PIQA, DLRP preserves near-full ranks across most layers, indicating limited tolerance for dimensionality reduction. In contrast, summarization tasks admit lower ranks: XSum maintains lower ranks than reasoning tasks but consistently higher than CNN/DM, while CNN/DM exhibits the lowest ranks overall. These patterns suggest that, depending on task characteristics, DLRP allocates more KV-cache capacity to the most consequential layers by assigning higher ranks, while reducing capacity in less influential layers via lower-rank projections. This behavior is conceptually aligned with the Lottery Ticket Hypothesis (LTH) (Frankle & Carbin, 2018). Rather than relying on the full KV-cache basis, DLRP identifies a *winning subspace*—a compact set of significant dimension—that, once fine-tuned, suffices to match the performance of the full model on the target tasks. The proposed regularizer and energy-based rank selection serve as a subspace-discovery mechanism, concentrating KV cache capacity on informative dimensions while suppressing uninformative ones. Unlike classical LTH, which searches for sparse subnetworks in weight space (often via pruning and rewinding), DLRP operates in the KV-cache space and achieves compression by learning a low-rank projector.

# F STATEMENT OF LARGE-LANGUAGE-MODEL (LLM) USAGE

The authors acknowledge that a large-language-model (LLM) was employed as a general-purpose assistance tool during the preparation of this manuscript. Specifically, the following tasks were supported by the LLM under the direct supervision of the authors:

- Formatting and LaTeX assistance – The LLM supplied LaTeX snippets for tables, equations, and figure captions. The authors integrated these snippets into the manuscript and performed all final compilation and formatting checks.
- Language polishing – The LLM was used to improve readability, correct grammar, and adjust stylistic tone across the entire manuscript. The final wording reflects the authors' own decisions after thorough review.

All content generated by the LLM was fully supervised, fact-checked, and substantially revised by the human authors before inclusion in the final version. No portion of the manuscript was submitted to the LLM for autonomous generation without subsequent author verification.

The authors affirm that the intellectual contributions, experimental design, data analysis, and conclusions are entirely their own work, and that the LLM served only as an auxiliary writing and editing aid.

