# OpenReview forum: "Deep Low Rank Projector for KV Cache Compression"
_ICLR.cc/2026/Conference — ICLR 2026 Conference Withdrawn Submission_

### Official Review · Reviewer_9119 · 2025-10-31

**Soundness:** 2
**Presentation:** 2
**Contribution:** 2
**Rating:** 2
**Confidence:** 5

**Summary:**

The paper proposes a Deep Low-Rank Projector (DLRP) to compress KV caches when models are fine-tuned with LoRA. It trains deep linear adapters with a Frobenius-norm surrogate to promote low rank, selects rank via energy thresholds, and folds the projector for inference. Experiments on Qwen and Mistral report better accuracy than applying token-eviction compression baselines on fine-tuned at similar compression ratios.

**Strengths:**

+ Clear, practical pipeline: train DLRP, energy-based rank selection, fold for inference.
+ Broad evaluation across models (Qwen/Mistral) and tasks with multiple compression rates.

**Weaknesses:**

+ **Novelty & positioning**: Trained low-rank projectors for KV compression have appeared recently (e.g., MatryoshkaKV); overlap and distinctions are not sufficiently discussed, weakening the originality claim.

+ **Baseline selection**: DLRP targets head-dimension redundancy, yet most baselines are token-eviction methods. Missing like-for-like baselines (e.g., EigenAttn [7] or Palu [4] with LoRA finetuning and other low-rank/train-to-compress methods) make it unclear whether gains stem from DLRP or from weaknesses of eviction on these tasks.

+ **Related work gaps**: Recent training-based KV compression exploring head-dimension redundancy (e.g., MHA2MLA [3], TransMLA[2]) are not adequately discussed, despite clear relevance to the paper’s “KV-compress + fine-tune” scope.

+ **Task choice**: Benchmarks are mostly short-context (e.g., PIQA, HellaSwag). Long-context stress tests (e.g., RULER [5], SCBench [6]) are lacking, leaving the real-world robustness under memory pressure unclear.

[1]. MatryoshkaKV: Adaptive KV Compression via Trainable Orthogonal Projection

[2]. TransMLA: Multi-Head Latent Attention Is All You Need

[3]. Towards Economical Inference: Enabling DeepSeek's Multi-Head Latent Attention in Any Transformer-based LLMs

[4]. Palu: Compressing KV-Cache with Low-Rank Projection

[5]. RULER: What's the Real Context Size of Your Long-Context Language Models?

[6] SCBench: A KV Cache-Centric Analysis of Long-Context Methods

[7]. Eigen Attention: Attention in Low-Rank Space for KV Cache Compression

**Questions:**

See the weaknesses.

---

### Official Review · Reviewer_i3sK · 2025-11-01

**Soundness:** 2
**Presentation:** 3
**Contribution:** 2
**Rating:** 2
**Confidence:** 3

**Summary:**

This paper identifies that PEFT-adapted LLMs (e.g., LoRA fine-tuned models) are significantly more sensitive to KV cache compression than base models, and proposes the Deep Low-Rank Projector (DLRP) to address this issue. Authors propose Deep Linear Networks as trainable adapters with a novel regularizer that approximates the nuclear norm to learn task-adaptive low-rank projections of the KV cache head dimension, automatically discovering appropriate compression levels through energy-based rank selection. Experiments across multiple models (Qwen3-4B/8B, Mistral-7B) and benchmarks show that the method substantially outperforms five baseline methods, especially at high compression rates (e.g., 59.4% vs 26.5% on GSM8K at 50% compression), while revealing interpretable patterns where reasoning tasks preserve higher ranks and summarization tasks adopt more aggressive compression.

**Strengths:**

- Evaluations are performed across multiple model sizes and different datasets with multiple compression rates.
- Practical and deployment friendly design. The linear matrices can be folded to have little to no overhead during deployment.
- Demonstrates interpretable behavior by automatically discovering task-appropriate compression levels: reasoning tasks preserve higher ranks while redundant long-context tasks (CNN/DM) achieve lower ranks

**Weaknesses:**

**Potential Error in the derivation of Theorem 3.1**
- In Equation (20), (21) the denominator should be N^N instead of N. While the stated theorem is still true as \frac{1}{N^N} <= \frac{1}{N}, the claimed bound becomes **exponentially loose** as depth increases.
- This did not seem to affect the results much, because authors only tried N=1,2,3 (from table3). However there is no theoretical backing anymore to use the said regularization loss. Once you accept arbitrarily loose bounds, the specific choice of \Sigma||D_n||_F becomes **unprincipled**. The authors need to argue why *their* loose bound is the right one to use.


**Motivation for DLN is lacking**
- What training dynamics of DLN motivates their use here? (line 202), “..induce highly non-convex training objectives…and exhibit an implicit bias towards low-rank solutions..”.  How do these help the task? If low-rankedness is required, why not use a projection and explicitly enforce it with a loss instead of an indirect method?
- With N=2, (table 3) the method essentially reduces to “finding the lowest ranked update on W and then fine tuning it with that rank”.
  - Determining that different tasks require updates of different ranks is not conceptually novel.
- Multiple LoRA methods exist that optimize the singular values of the weights ([e.g.](https://arxiv.org/abs/2405.19597)) which need to be discussed and compared with.
- If alpha=0 works, was the regularization even needed? Does having higher alpha, reduce the selected rank?

**Unfair Evaluation Setup**
- Line 316, “..first finetuned with LoRA (Hu et al (2022)) (rank 32)..” are all models finetuned with rank 32 irrespective of compression?
- Line 323, “..Ranks are selected using energy thresholds that correspond approximately to 10%, 25%, 50%..”. With different ranks to LoRA and the DLRP, the proposed method uses more task-dependent parameters for each dataset. Which makes the comparison unfair.

**Questions:**

See weaknesses

---

### Official Review · Reviewer_mc12 · 2025-11-01

**Soundness:** 2
**Presentation:** 3
**Contribution:** 2
**Rating:** 4
**Confidence:** 4

**Summary:**

This paper introduces Deep Low-Rank Projector (DLRP), an adapter module designed to compress the Key-Value (KV) cache of large language models (LLMs), particularly in parameter-efficient fine-tuning (PEFT) settings such as LoRA. The key innovation lies in leveraging Deep Linear Networks (DLNs) to model the projection process and introducing a nuclear-norm-inspired regularizer to promote low-rank structure without performing expensive SVD. After training, the method selects the smallest energy-preserving rank, constructs the DLRP, and fine-tunes it on downstream tasks. Experiments on Qwen3-4B, Qwen3-8B, and Mistral-7B show that DLRP significantly outperforms existing methods such as SnapKV, StreamingLLM, PyramidKV, and KeyDiff, especially in PEFT settings.

**Strengths:**

1. Novel formulation:
The use of Deep Linear Networks for cache compression is novel and theoretically grounded. The proposed regularizer offers an elegant and computationally efficient surrogate to the nuclear norm.
2. Comprehensive experiments:
Evaluation across multiple benchmarks (GSM8K, PIQA, HellaSwag, XSum, CNN/DM) and models (Qwen, Mistral) demonstrates strong empirical robustness.
3. Clear motivation:
The paper clearly identifies and addresses the overlooked interaction between KV-cache compression and PEFT.

**Weaknesses:**

1. Generalization beyond a single task or dataset:
It appears that the DLRP needs to be trained separately for each downstream task, since the rank-selection step is based on task-specific energy thresholds and the adapter is fine-tuned afterward.
Question: Does this imply that if we want to apply DLRP to another downstream task, we must re-train the projector from scratch?
If so, this may limit scalability and increase computational overhead. I suggest the authors explore task-agnostic training or transferability across related domains.
2. Inference-time efficiency vs. compression ratio:
While the paper demonstrates strong accuracy retention, it does not explicitly quantify inference latency reduction.
Question: As the compression ratio increases, does the actual inference time decrease proportionally?
Given that the DLRP folds into a single linear map, it would be useful to report speedups or GPU memory usage during real-time decoding.
3. Integration with other compression paradigms:
DLRP currently focuses on the head-dimension compression only. The authors acknowledge this in the Limitations section.
→ Suggestion: It would be interesting to discuss or experiment with combining DLRP with sequence-length compression methods (e.g., StreamingLLM, H2O) or quantization-based approaches (e.g., bit-level cache compression). Such hybrid methods may yield higher compression without retraining.

**Questions:**

See weaknesses

---

### Official Review · Reviewer_ravY · 2025-11-01

**Soundness:** 3
**Presentation:** 2
**Contribution:** 2
**Rating:** 4
**Confidence:** 3

**Summary:**

DLRP is a PEFT-friendly KV-cache compressor that reduces the head dimension via trained Deep Linear Projectors with an SVD-free low-rank regularizer. After selecting an energy-preserving rank, the adapters are fine-tuned and folded into single linear maps for inference, consistently beating prior methods (SnapKV, StreamingLLM, Knorm, PyramidKV, KeyDiff) at 10/25/50% compression on Qwen3-4B/8B and Mistral-7B, with task-/layer-adaptive ranks that keep reasoning accuracy while exploiting redundancy in long-context summarization.

**Strengths:**

- First KV-cache compression method explicitly tailored for LoRA- or PEFT-adapted models, addressing their higher sensitivity to compression.
- Introduces Deep Low-Rank Projector (DLRP) built on deep linear networks, enabling effective compression along the head dimension.
- Proposes an SVD-free nuclear-norm surrogate (Frobenius-norm–based) that promotes low-rank structure with much lower computational cost.

**Weaknesses:**

- DLRP only compresses the head dimension, leaving other major axes (layers, sequence length, number of heads) untouched—limiting overall memory savings.
- The two-stage process (regularized DLP training + DLRP fine-tuning) increases training cost and implementation overhead compared to simpler post-hoc compression methods.
- The paper focuses on accuracy retention but omits decoding speed or latency improvements, which are critical for practical deployment
- Performance depends on choices like the energy threshold and regularization strength (α), yet guidelines for tuning are minimal.

**Questions:**

How well does DLRP perform on non-instruction-tuned or multilingual models? Since the experiments focus on English datasets, it would be useful to evaluate whether the learned low-rank subspace transfers across domains and languages.

---

### Note · Authors · 2025-11-24

I have read and agree with the venue's withdrawal policy on behalf of myself and my co-authors.